# Multiple Factors Influence the Incubation Period of ALS Prion-like Transmission in SOD1 Transgenic Mice

**DOI:** 10.3390/v15091819

**Published:** 2023-08-26

**Authors:** Jacob I. Ayers, Guilian Xu, Qing Lu, Kristy Dillon, Susan Fromholt, David R. Borchelt

**Affiliations:** 1Institute for Neurodegenerative Disease, University of California, San Francisco, CA 94158, USA; 2Department of Neuroscience, Center for Translational Research in Neurodegenerative Disease, University of Florida, Gainesville, FL 32610, USA; xugl@ufl.edu (G.X.);; 3SantaFe HealthCare Alzheimer’s Disease Research Center, McKnight Brain Institute, University of Florida, Gainesville, FL 32610, USA

**Keywords:** amyotrophic lateral sclerosis, superoxide dismutase-1, prion, seeding

## Abstract

Mutations in superoxide dismutase 1 (SOD1) that are associated with amyotrophic lateral sclerosis (ALS) cause its misfolding and aggregation. Prior studies have demonstrated that the misfolded conformation of ALS-SOD1 can template with naïve SOD1 “host proteins” to propagate, spread, and induce paralysis in SOD1 transgenic mice. These observations have advanced the argument that SOD1 is a host protein for an ALS conformer that is prion-like and experimentally transmissible. Here, we investigated the propagation of different isolates of G93A-SOD1 ALS conformers using a paradigm involving transmission to mice expressing human G85R-SOD1 fused to yellow fluorescent protein (G85R-SOD1:YFP). In these studies, we also utilized a newly developed line of mice in which the G85R-SOD1:YFP construct was flanked by loxp sites, allowing its temporal and spatial regulation. We used methods in which the G93A ALS conformers were injected into the sciatic nerve or hindlimb muscle of adult transgenic mice. We observed that the incubation period to paralysis varied significantly depending upon the source of inoculum containing misfolded G93A SOD1. Serial passage and selection produced stable isolates of G93A ALS conformers that exhibited a defined minimum incubation period of ~2.5 months when injected into the sciatic nerve of young adult mice. As expected, neuronal excision of the transgene in loxpG85R-SOD1:YFP mice blocked induction of paralysis by transmission of G93A ALS conformers. Our findings indicate that G93A ALS conformers capable of inducing disease require neuronal expression of a receptive host SOD1 protein for propagation, with a defined incubation period to paralysis.

## 1. Introduction

ALS is a fatal neurodegenerative disease in which both upper and lower motor neurons degenerate to cause systemic paralysis [1]. The way that ALS progresses in patients, by the apparent spread of weakness from one muscle group to another, presents parallels to prion disease where misfolded prion proteins spread along neuroanatomical pathways to engulf the central nervous system [1]. Approximately 10% of familial ALS cases are caused by mutations in the gene encoding the antioxidant enzyme *SOD1* and display the same clinical presentation of a spreading disease [2,3]. To date, more than 170 mutations at 80 different amino acid positions in SOD1 have been associated with ALS (https://alsod.uk accessed on 1 July 2023). The vast majority of disease-associated mutations are mis-sense point mutations. Although SOD1 is broadly expressed throughout the body (https://www.proteinatlas.org accessed on 1 July 2023), mutant SOD1 is selectively toxic to motor neurons, causing degeneration [2]. At autopsy, the surviving motor neurons in the spinal cords of SOD1-ALS patients contain inclusions that are immunoreactive to SOD1 antibodies [4,5]. Similar pathologies have been described in various transgenic mice that over-express human SOD1 with mutations associated with ALS [6,7,8,9]. A study of 27 disease variants of human SOD1 in cell culture over-expression revealed that the ALS mutations cause conformational changes that induce the protein to misfold and self-associate into detergent-insoluble aggregates [10]. Thus, one common feature of ALS caused by mutations in SOD1 is that the mutations induce the misfolding and aggregation of the SOD1 protein (reviewed by [11,12]).

One remarkable feature of SOD1-ALS is that the duration of disease tends to be similar among patients with the same mutation. For example, individuals that inherit the A4V or G93A variants survive 1.5–2.5 years (average) after onset, whereas individuals inheriting the G37R or H46R variant may live up to 17 years [10]. These distinctive survival characteristics of disease in individuals with different mutations suggests that attributes of the misfolded SOD1 proteins, by some means, dictate the rate of disease progression. We have been interested in whether SOD1 mutations and specific conformations of the protein could affect their prion-like activity and explain why different mutations exhibit different durations in patients [13,14].

Studies in cell culture models were the first to establish the potential for misfolded SOD1 to propagate between cells [15,16,17]. Evidence for in vivo propagation has been obtained by injecting spinal cord homogenates prepared from paralyzed mutant SOD1 transgenic mice or human SOD1-ALS tissues into the spinal cords of young mice expressing the G85R variant of human SOD1 [18,19,20]. We have extensively used a version of G85R-SOD1 mice in which the human protein is fused in frame to yellow fluorescent protein (YFP) [21]. Mice that are heterozygous for the transgene do not develop paralysis until after 18 months of age [21,22], whereas newborn mice injected with preparations containing misfolded SOD1 conformers develop paralysis in less than 3 months [18,22]. Additionally, we observed that injecting fibrilized recombinant human SOD1 can induce early-onset paralysis in G85R-SOD1:YFP mice [14,23]. An important consideration in the prion-like propagation of misfolded SOD1 is the interaction of the seeding ALS conformer with the SOD1 expressed by the host. Mice that express the G85R variant of human SOD1 are highly permissive hosts, whereas mice that express wild-type SOD1, or mice that express ALS variants that are biochemically similar to wild-type SOD1, show little or no propagation of ALS conformers [14,18,22]. Because of these limitations in host susceptibility, we refrain from referring to propagating SOD1 as an ALS prion, in favor of the term ALS conformer.

The ability to induce early-onset ALS by injecting misfolded seeds of mutant SOD1 allowed us to develop a paradigm in which spinal tissue homogenates were injected into the sciatic nerve of young G85R-SOD1:YFP mice [13]. In our initial studies, we observed that injection of tissue homogenates from a paralyzed animal into the sciatic nerve of a young G85R-SOD1:YFP animal produced a rapidly progressing disease in which the ipsilateral limb became weak and paralyzed in 2–3 months post-injection (p.i.). The disease spread to the contralateral limb, which became paralyzed 1–2 weeks later [13]. At one month p.i., we observed that the injected G93A ALS conformers induced low levels of inclusion pathology in the ipsilateral dorsal root ganglion. By 2 months p.i., inclusions were evident throughout the lumbar spinal cord with pathology also evident in anatomically connected nuclei of the brain (reticular formation, lateral vestibular nucleus, and red nucleus) [13]. As weakness engulfed both hindlimbs, inclusion pathology became more abundant in the spinal cord and began to appear in additional brain structures. In this original study, we used a protocol in which we first induced disease in G85R-SOD1:YFP mice by injecting spinal tissue from a paralyzed G93A animal into the spinal cord of newborn G85R-SOD1:YFP animals. When one of these animals developed paralysis 5.5 months later, tissue was harvested and used to prepare spinal tissue homogenates for injection into the sciatic nerve of naïve G85R-SOD1:YFP mice. From these original studies, we concluded that G93A ALS conformers could be propagated along anatomical pathways to produce paralytic disease.

In the present study, we have extended our examination of the characteristics of ALS induction in mice injected with different types of preparations containing G93A SOD1 ALS conformers. To model the spreading pathology that seems to occur in human ALS, we injected inoculum into the sciatic nerve of young adult G85R-SOD1:YFP mice. Our findings show that the incubation period to paralysis induced by G93A ALS conformers is influenced by the source of inoculum. Recombinant G93A ALS conformers showed much longer incubation periods than G93A conformers arising in transgenic mice. By serial passage and selection, we isolated preparations containing G93A ALS conformers that consistently induced paralysis by 2.5–4 months post-inoculation. Surprisingly, this incubation period was similar to the minimum incubation period we had observed for G93A ALS conformers in G85R-SOD1:YFP mice that were injected intraspinally as newborns [22]. Using mice that express G85R-SOD1:YFP from transgenes encoding loxp sites, we show that neuronal expression is critical for the propagation of paralysis-inducing G93A ALS conformers. Our findings indicate that G93A ALS conformers propagating primarily in neurons cause paralysis and exhibit a prion-like minimum incubation period.

## 2. Methods

### 2.1. Animal Models

The G85R-SOD1:YFP mice used in this study were a kind gift from Drs. Jiou Wang and Arthur Horwich [21], and were maintained on the FVB/NJ background. These mice express the transgene in skin and can be identified via illumination and specialized filter goggles (BLS Ltd. Budapest, Hungary). The loxpG85R-SOD1:YFP mice were generated by mutagenesis of a ~10 kb fragment of genomic human DNA in which exons 3, 4, and 5 had been fused to eliminate intervening intronic sequences [24]. The plasmid backbone for this construct was pBR322, which was used in the original cloning of the human genomic DNA fragment containing the *SOD1* gene [25]. PCR-based mutagenesis was used to introduce the G85R mutation, loxp sites, and cDNA for YFP (Appendix A). The YFP open reading frame was inserted at the c-terminus of SOD1 such that the normal stop codon was eliminated to produce a continuous fusion protein of SOD1:YFP. For identification of genotype, DNA was extracted from mouse tail biopsies and analyzed via PCR using the following primers:

Hu-S—TCA AGC GAT TCT CCT GCC T (GRCh38/hg38) chr21 (+) strand 31,666,068–31,666,087; intronic sequence 5′ to exon 3.

Mo-S—TAC ATA TAG GGG TTT ACT TCA T GRCm38/mm10 90,224,306–90,224,326; intronic sequence 5′ to exon 3.

H/M-AS—CAC ATT GCC CAR GTC TCC A (R = A/G) GRCm38/mm10 (−) strand 90,225,151–90,225,169 and GRCh38/hg38 (−) strand 31,667,264–31,667,282; conserved sequence in exon 4.

The PCR product amplified with Hu-S and H/M-AS primers is ~500 bp, whereas the product amplified with Mo-S and H/M-AS is ~900 bp.

All studies involving mice were approved by the Institutional Animal Care and Use Committee (IACUC) at the University of Florida in accordance with all state and federal guidelines. All animals were housed one to five to a cage and maintained on ad libitum food and water with a 14 h light and 10 h dark cycle.

### 2.2. Preparation of Inoculum

Spinal cord tissues were weighed and homogenized in ten volumes of PBS to produce a 10% homogenate (*w*/*v*) containing 1:100 *v/v* protease inhibitor cocktail (Sigma, St. Louis, MO, USA), as previously described [18]. Homogenization was accomplished by sonicating spinal tissues with a probe sonicator 4 times for 20 s each, with cooling on ice between bouts of sonication. The crude homogenates were then clarified using a low-speed spin at ~800× *g* for 10 min and the supernatants were aliquoted to produce inoculum, which was stored at −80 °C. Repeated freeze/thaw of inoculum was avoided. We also generated concentrated inoculum via centrifugation at ~100,000× *g* for 30 min at 4 °C in an ultracentrifuge. The supernatant was discarded and the pellet was resuspended in one-fifth of starting volume of PBS with protease inhibitor via sonication (probe sonicator applied to the outside of the tube). The suspension was then aliquoted into 10 uL aliquots in PCR tubes and stored at −80 °C.

Two preparations of pooled inocula were generated by combining spinal cord tissue homogenates from different animals. The initial pool was generated by selecting three banked frozen spinal cords from paralyzed G85R-SOD1:YFP mice that had been seeded by P0 intraspinal (i.sc.) injection. The incubation periods for these three animals were 2.8, 7.9, and 12 months p.i. The second pool was generated after serial passage by P0 i.sc. injection. The first passage involved seeding with spinal homogenates from a paralyzed G93A mouse to induce paralysis. Tissues from a G85R-SOD1:YFP animal that became paralyzed at 2.6 months were then used in second passage to naïve G85R-SOD1:YFP mice (P0 i.sc. injection). Frozen spinal cord tissues from 4 animals that received these injections were selected to produce a second pool. The incubation periods for the 4 animals were 2.3, 2.4., 2.4, and 2.5 months p.i. Pooled inocula were stored in the same manner as described above.

### 2.3. Sciatic Nerve Injections

The method used for sciatic nerve injections has been described in detail previously [13,22], with the following modification. To reduce postoperative pain, after the injection site was shaved and sterilized, the site was injected subcutaneously with 100 ul of a 0.05% solution of bupivacaine. After a few minutes, the initial incision was made to expose the sciatic nerve and inoculum was injected as described [13,22].

### 2.4. Euthanasia and Tissue Collection

Mice were deeply anesthetized with isoflurane and then perfused transcardially with 20 mL of PBS to cause death. The spinal columns and brains were immediately removed. The brains were bisected sagittally and one hemisphere was drop-fixed in 4% paraformaldehyde in PBS for ~48 h at 4 °C and the other hemisphere was flash-frozen on dry ice and then stored at −80 °C. The spinal columns were removed and cut into 4 sections. The cervical and lumbar segments were drop-fixed in 4% paraformaldehyde in PBS for ~48 h at 4 °C. Spinal cord tissues were dissected from the other segments and flash-frozen on dry ice before storage at −80 °C.

### 2.5. Neuropathology

For imaging YFP fluorescence, paraffin-embedded tissues were cut at 5 μm and then deparaffinized and coverslipped with water. Fluorescence images were visualized with epifluorescence Olympus BX60 microscope. After images were captured, the coverslip was removed and the section was stained using Campbell–Switzer silver staining as previously described [22]. Silver-stained slides were imaged in an Aperio Scanscope XT (Lecia Biosystems, Deer Park, IL, USA) at 20×. Images were digitally magnified in ImageScope to produce the representative images. The extracted images obtained from ImageScope were processed using GIMP (GNU Image Manipulation Program; version 2.10.34) software, which involved precise cropping of the images and addition of scale bars using the pencil tool, with the length of the bars measured in pixel counts.

### 2.6. Statistical Analysis

Differences in survival among the cohorts of Line 230 mice injected with inoculum were determined using GraphPad Prism 9.5.1 (GraphPad Software, Boston, MA, USA) software employing the log-rank (Mantel–Cox) test. *p* values less than 0.05 were considered significant.

## 3. Results

In previous studies, we have demonstrated that spinal cord tissue homogenates from symptomatic G93A transgenic mice can induce accelerated paralysis in G85R-SOD1:YFP mice through exogenous administration [18,22]. Initially, we injected tissue homogenates containing misfolded mutant SOD1 into the spinal cords of newborn P0 G85R-SOD1:YFP mice. We then identified mice that showed accelerated motor neuron disease (MND) and prepared new tissue homogenates to serve as seeds. By this method, we assay whether a specific conformation from the original mutant SOD1 can be transmitted to G85R-SOD1:YFP mice and propagated as a distinct “strain” of misfolded SOD1 [14]. Our nomenclature identifies the inocula, using the terminology P1 to designate that the source was a “first-passage” recipient of seeds and to designate the mutation site of the seed source, and the superscript G85R-SOD1:YFP designates the host for the first passage (e.g., P1-G93A^G85R-SOD1:YFP^). We have shown that sciatic nerve injection of spinal homogenates from a P1-G93A^G85R-SOD1:YFP^ animal induced a rapidly progressing MND in naïve G85R-SOD1:YFP mice [13]. By 3 months of age, 8 of the 10 injected mice had developed bilateral hindlimb paralysis and met criteria for euthanasia (data reproduced from [13] in Figure 1 for convenience, Columns 1 and 2). The P1-G93A^G85R-SOD1:YFP^ inoculum used for injection was derived from a single animal that had developed paralysis at ~5.5 months of age (Figure 1, Column 1 and reproduced from [13]). This same inoculum had also been used in intraspinal injections of newborn G85R-SOD1:YFP mice, producing paralysis in 8 of 8 injected mice at an average incubation period of 2.8 months [14].

To guard against animal-to-animal variation in strains, we moved to a paradigm in which homogenates from 3 animals were pooled. We prepared a pooled inoculum from three P1-G93A^G85R-SOD1:YFP^ mice which had developed paralysis at 2.8, 7.9, and 12 months of age (Figure 1, Column 3; Appendix A). In the newborn injections that were used to produce the first-passage animals, we presumed inoculum was widely distributed throughout the CNS based on prior experience with similar injections of recombinant adeno-associated viruses [26]. Thus, in this paradigm, the age to paralysis was not thought to be related to a rate of spread, but rather an initial efficacy of delivery. Pathologically, all three of these animals displayed abundant inclusions, including the animal with the longest incubation period (Appendix A). Thus, we expected that the spinal tissues from these mice would have equivalent seeding potential. Surprisingly, the performance of the pooled P1-G93A^G85R-SOD1:YFP^ inoculum was quite different from the original observations made with inoculum from a single animal. Only 4 of 11 mice that were injected with the pooled inoculum developed bilateral hindlimb paralysis before 16.5 months p.i., and the intervals to paralysis p.i. ranged from 7 to 11 months (Figure 1, Columns 3 and 4). As expected, the lumbar spinal cords from all of the paralyzed mice exhibited some level of G85R-SOD1:YFP inclusion pathology (Figure 2). In some of the older animals, we observed lipofuscin accumulation that intermingled with G85R-SOD1:YFP fluorescence (Appendix A). We used two different methods to distinguish lipofuscin from the G85R-SOD1:YFP inclusions. One method was to subtract any fluorescence from the red channel that originated from lipofuscin (Appendix A). The other method used was Campbell–Switzer silver staining to confirm the presence of inclusions, as we have previously described [22] (Figure 2D–F). Using these methods, we confirmed that spinal cord tissues from paralyzed G85R-SOD1:YFP mice exhibited inclusion pathology. Together, these data suggested that pooling homogenates from animals with different incubation periods diminished the consistency of the inoculum.

### 3.1. Strains of Recombinant Fibrilized G93A SOD1

We have recently shown that recombinant G93A SOD1 fibrils induce accelerated paralysis when injected into the spinal cords of newborn G85R-SOD1:YFP mice [23]. The incubation period to paralysis for these mice ranged from 8.0 to 15 months p.i. We then asked whether we could reproduce the rapidly spreading strain of G93A-SOD1-ALS conformers that we initially observed [13]. We prepared inoculum from a P1-recG93A^G85R-SOD1:YFP^ animal that developed paralysis at 9.3 months p.i., which in this case was the shortest incubation period (Figure 1, Column 5; Appendix A). When this inoculum was injected into the sciatic nerve of 2-month-old naïve G85R-SOD1:YFP mice, we observed highly variable incubation periods, with an average that was much longer than the 3-month average incubation period we observed in our original index cohort (Figure 1; compare Columns 2 and 6). As expected, paralyzed mice exhibited abundant inclusion pathology (Figure 3). The longer incubation period for passaged recG93A conformers suggested that the strain, or strains, created by fibrilizing recombinant G93A SOD1 was not identical to the strain produced by seeding with spinal homogenates from G93A mice, indicating that the context in which the G93A-SOD1-ALS conformers arise may influence their strain properties.

### 3.2. Neuronal Expression of G85R-SOD1:YFP Mediates Susceptibility to Paralysis via Propagation of G93A ALS Conformers

SOD1 is broadly expressed in the CNS in multiple cell types [27] {https://www.proteinatlas.org accessed on 1 July 2023}. To begin to examine which cell types in the CNS may participate in the propagation of SOD1 ALS conformers, we developed a line of G85R-SOD1:YFP mice in which the transgene construct had been modified to encode loxp recombination sites that would delete coding sequences when exposed to Cre recombinase (Figure 4A). The starting construct for this transgene was a version of genomic human DNA in which exons 3, 4, and 5 had been fused to eliminate introns 3 and 4 [24]. We introduced the G85R mutation and inserted cDNA for YFP in frame while eliminating the normal stop codon for SOD1. One loxp site was inserted at the 5′ end of the gene and the other was inserted at the 3′ end of the coding sequence for YFP. Two lines of mice were identified that expressed the gene at levels similar to the original G85R-SOD1:YFP mice (Lines 216 and 230). We were able to make Line 230 mice homozygous for the transgene, and all further experimentation focused on this line. Heterozygous Line 230 mice express G85R-SOD1:YFP at levels similar to the original G85R-SOD1:YFP mice developed by Wang and colleagues [21] (Figure 4B, lanes 2 and 3). Homozygous Line 230 mice were viable and fertile and expressed the transgene (Figure 4B, lane 4). When homozygous Line 230 mice were crossed to mice expressing Cre recombinase using a synapsin promoter (Jax stock no: 003966 [28]), the resulting offspring that were bigenic for Cre and G85R-SOD1:YFP had much lower expression of the transgene in spinal cord lysates (Figure 4B, lanes 5 and 6). These data demonstrate that neurons are the primary cell type that accumulates G85R-SOD1:YFP in this new line of loxpG85R-SOD1:YFP mice (Line 230). This finding is comparable to the original description of G85R-SOD1:YFP mice [21].

To examine SOD1 seeding and propagation in this new model, we challenged the mice via sciatic nerve injection with seeds derived from G93A ALS conformers. In preparation for these studies, we first sought to generate an inoculum of G93A ALS conformers that produced a consistent incubation period to paralysis. We started with tissue from a G85R-SOD1:YFP mouse that was induced to develop paralysis by 2.5 months (Appendix A) and seeded multiple naïve G85R-SOD1:YFP mice, identifying four animals that showed a similar ~2.5-month incubation period (Figure 5, Column 1, gray circles). Spinal cord tissues from these four mice were homogenized, pooled, and aliquoted to generate a pooled inoculum that was used for sciatic nerve injection of homozygous Line 230 mice. This inoculum was effective in inducing paralysis in the Line 230 mice, with most of the mice developing paralysis in less than 4 months p.i. (Figure 5, Column 2). Notably, there was still some variability in incubation periods, with three animals showing a later age to paralysis and one animal showing no symptoms or pathology when euthanized at 10 months of age. To determine whether the incubation period was related to the amount of seed, or titer, in the inoculum, we prepared a more concentrated homogenate of aggregated G85R-SOD1:YFP seeds through centrifugation (see Methods and [22]). When these preparations were used for sciatic nerve injection in homozygous Line 230 mice, we continued to observe a minimum incubation period of ~2.5 months. Although the incubation periods tended to be slightly more consistent, we still observed inexplicable outliers and there was no statistical difference in survival (Figure 5, Column 3; Appendix A). As expected, paralyzed Line 230 mice were found to have abundant inclusion pathology in lumbar spinal cord tissues (Figure 6; Appendix A). Collectively, these data indicate that serial passage and selection of early-onset responder mice produced an inoculum of G93A ALS conformers that was more consistent and induced paralysis as early as 2.5 months p.i.

To determine whether another route of peripheral inoculation could eliminate the outliers, we injected the concentrated inoculum into the muscle of young adult homozygous Line 230 mice. Similar to what has recently been reported by Keskin [31], muscle injection was ineffective in transmission of SOD1-ALS conformers (Figure 5, Column 4). A subset of muscle-injected mice developed paralysis, but the ages at which disease developed were not statistically different from those of uninjected homozygous Line 230 mice (Figure 5, Column 5; Appendix A). Lumbar spinal tissues from asymptomatic mice that were inoculated in the muscle did not exhibit inclusion pathology, whereas the pathology burden in paralyzed muscle-injected mice was similar to that of age-matched uninjected mice that spontaneously developed paralysis (Appendix A).

To confirm the importance of neuronal expression in the propagation of paralysis-inducing ALS conformers, we crossed homozygous Line 230 mice to Syn-Cre mice and injected the sciatic nerve of adult (~2.5-month-old) mice with the same pooled G93A inoculum described in Figure 5. For these experiments, the inoculum was not concentrated. Six of eleven mice that were negative for Cre developed paralysis between 2.3 and 4.2 months p.i. (Figure 6). Of the five mice in this group that remain alive at the time of writing, two mice were 9.5 months p.i. with three additional mice at 11 months p.i. (Figure 7, open circles). At the time of writing, none of the 11 mice that were positive for Cre had developed paralysis (9.5–11 months p.i.; Figure 7). The difference in survival of these two groups reached statistical significance (*p* = 0.0051, log-rank [Mantel–Cox] test). Notably, the efficiency of seed transmission in heterozygous Line 230 mice was substantially lower than that in the homozygous mice, indicating that expression levels of G85R-SOD1 in the host are another factor that influences the propagation of ALS conformers in SOD1 transgenic mice. As expected, mice that developed paralysis also developed substantial inclusion pathology (Figure 7B,C). Though it is not surprising that elimination of neuronal expression of G85R-SOD1:YFP would inhibit induction of motor neuron disease in this model, the data provide a confirmation of the role of neurons in propagating misfolded SOD1 ALS conformers that induce paralysis.

## 4. Discussion

We have examined the propagation of different isolates of G93A-SOD1 ALS conformers, using a paradigm in which G93A SOD1-ALS conformers were transmitted to mice expressing human G85R-SOD1 fused to yellow fluorescent protein (G85R-SOD1:YFP). The G93A ALS conformers were injected into the sciatic nerve or hindlimb muscle of adult mice, and we assessed the incubation period to paralysis. Depending upon the source of inoculum containing misfolded G93A SOD1, the incubation periods varied significantly. Notably, the incubation period for ALS conformers derived from recombinant G93A fibrils was substantially longer than what was observed for conformers derived from paralyzed G93A mice. Serial passage and selection of spinal cord-derived G93A ALS conformers produced inoculum that exhibited a defined minimum incubation period of ~2.5 months when injected into the sciatic nerve of adult mice. Interestingly, the minimum incubation period in G85R-SOD1:YFP mice injected as newborns was also ~2.5 months [22]. As expected, neuronal excision of the transgene in loxpG85R-SOD1:YFP mice blocked induction of paralysis following injection with G93A ALS conformers. Our findings indicate that G93A ALS conformers require neuronal expression of a receptive host SOD1 protein to induce paralysis, and that the minimum incubation period for induction of paralysis is ~2.5 months.

One of the most surprising findings was the performance of pooled homogenates from P1-G93A^G85R-SOD1:YFP^ mice. Using tissues from three animals that developed paralysis at different ages, but which all had high levels of inclusion pathology at the end stage (see Appendix A), we produced a pooled P1-G93A^G85R-SOD1:YFP^ inoculum. This inoculum was relatively inefficient at seeding and resulted in long incubation periods. These findings could be an indication that the three animals used to make this initial pool contained distinct substrains of G93A ALS conformers that in some manner diminished overall transmissibility when mixed. Notably, it appeared in our subsequent studies that serial passage and pooling of homogenates from animals with more similar incubation periods produced a more consistent inoculum.

To determine whether the sequence of mutant SOD1 was the sole determinant in incubation periods, we assessed the induction of paralysis by inoculum prepared from G85R-SOD1:YFP mice that had been injected with recombinant G93A fibrils. Although this inoculum induced paralysis at a high frequency (9 of 12 mice injected; see [22]), the incubation period was variable and much longer than the ~2.5-month incubation periods observed with the passage of the G93A ALS conformer derived from paralyzed G93A mice. It is possible that additional rounds of passage and selection would result in more congruent data. At minimum, our findings indicate that recombinant proteins fibrilized in vitro do not exhibit the same strain-like attributes of G93A ALS conformers that arise in tissues after serial passage. Notably, differences between recombinant strains and natural strains has been observed when comparing the cryo-EM structures of tau and synuclein fibrils purified from patient brains versus those produced using recombinant protein [32,33,34].

One question that is difficult to resolve is whether a technical aspect of this study explains some of the variability in outcomes. The sciatic nerve is a relatively small target and thus it is difficult to be assured of equivalent delivery of the inoculum in each animal. Notably, concentrating the inoculum by 5-fold did not reduce the minimum incubation period below ~2.5 months and did not fully alleviate variability. It is possible that variation in the amount of seed in the inoculum that penetrates the nerve may be responsible for some of the variability we observed. Alternatively, as noted above, it is possible that the inoculum used for some of these injections was a mixture of strains rather than a uniform preparation.

SOD1 is broadly expressed in the CNS and throughout the body [27] {https://www.proteinatlas.org accessed on 15 August 2023}. Within the CNS and spinal cord, all cell types express SOD1, but the levels are variable and the contribution of mutant SOD1 pathology in different cell types is similarly variable [27,35,36,37]. In SOD1 mouse models, mutant SOD1 inclusion pathology is abundant in neuronal cell bodies and within the neuropil, with limited pathology in astrocytes [8,38,39]. Inclusion pathology in the neuropil is difficult to assign to a particular cell origin, and thus the role of non-neuronal cells in propagating misfolded SOD1 is uncertain.

To examine which cell types in the CNS may participate in the propagation of paralysis-inducing SOD1 ALS conformers, we developed a new line of G85R-SOD1:YFP mice in which the transgene is flanked by loxp sites. Our construct was modeled after a similar construct for untagged G85R-SOD1 described by Wang and colleagues [40]. In the course of study to validate the utility of this line of mice in SOD1 ALS conformer transmission, we tested whether the transgene could be excised by coexpression of Cre recombinase via a synapsin promoter [28]. To avoid potential germline excision of the transgene, we crossed Syn-Cre mice to homozygous Line 230 mice so that all offspring would be F1 [41]. The original description of G85R-SOD1:YFP mice reported that transgene protein and mRNA are most abundant in neurons [21]. Our modified variant of G85R-SOD1:YFP, which added loxp sites and removed two introns, appears to accumulate primarily in neurons based on the absence of transgene-derived protein in Line 230 mice that are positive for Synapsin Cre. In a prior study of sciatic nerve-injected mice, we noted that the first detection of induced SOD1 misfolding was in cell bodies of sensory neurons in the ipsilateral dorsal root ganglion [13]. Synapsin is expressed broadly in all neurons of the brain and spinal cord (Allen Brain Atlas), and would be expected to eliminate expression in both sensory and motor neurons. Our findings show that deletion of the G85R-SOD1:YFP transgene in neurons lowers the susceptibility of the host to induction of paralysis by injecting G93A-ALS conformers. Future studies with Cre mice that express in more selected subsets of neurons could reveal the relative contribution of different neuronal populations to the spreading propagation of SOD1 ALS conformers in these mouse models.

## 5. Conclusions

In the present study, we have examined factors that influence the consistency of the prion-like spread of misfolded SOD1 in transgenic mouse models. We have shown that the source of the inoculum is a significant factor. We obtained the most consistent and reproducible results when we pooled spinal cords from mice that developed paralysis at roughly equivalent incubation periods. Interestingly, we observed that the minimum incubation time for G93A ALS conformers injected within the sciatic nerve was approximately 2.5 months, similar to what we previously observed in intraspinal injection of newborn G85R-SOD1:YFP mice [22]. This finding implies that the two routes of inoculation introduce the seeds to target neurons with equivalent efficiency and that the incubation period from inoculation to paralysis may not be a function of time to spread, but a function of the amount of time required for “infected” neurons to accumulate toxic conformers that mediate paralysis.

## Figures and Tables

**Figure 1 viruses-15-01819-f001:**
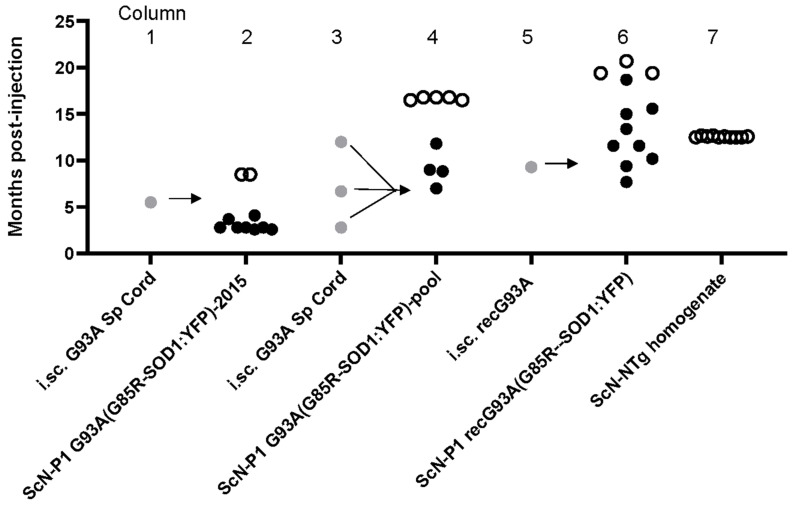
The source of G93A SOD1 ALS conformers influences seeding efficiency and incubation period. The scatter plot graphs survival data for G85R-SOD1:YFP mice inoculated with different preparations of tissue homogenates. The gray symbols note the incubation periods of animals that were selected from banked tissue as sources for preparation of inoculum. These mice were inoculated as newborns through intraspinal (i.sc.) injection (first passage) using spinal cord homogenates from a paralyzed G93A mouse or recombinant fibrils of mutant SOD1 (see [18,22]). The black symbols are animals that were inoculated through sciatic nerve (ScN) injection with spinal homogenates from the first-passage recipients. Columns 1 and 2 show data that are reproduced from [13] for convenience of comparison. Column 3 shows the incubation periods to paralysis for 3 animals that were used to make a pooled inoculum that was then injected into the sciatic nerve of recipient G85R-SOD1:YFP mice (Column 4). Column 5 indicates the incubation period for an animal injected with recombinant G93A fibrils; this animal produced inoculum for sciatic nerve injection (Column 6). Column 7 shows data for a cohort of sciatic nerve-injected G85R-SOD1:YFP mice that were inoculated with spinal homogenate from an asymptomatic nontransgenic mouse. All of these mice were asymptomatic when harvested at 12 months of age.

**Figure 2 viruses-15-01819-f002:**
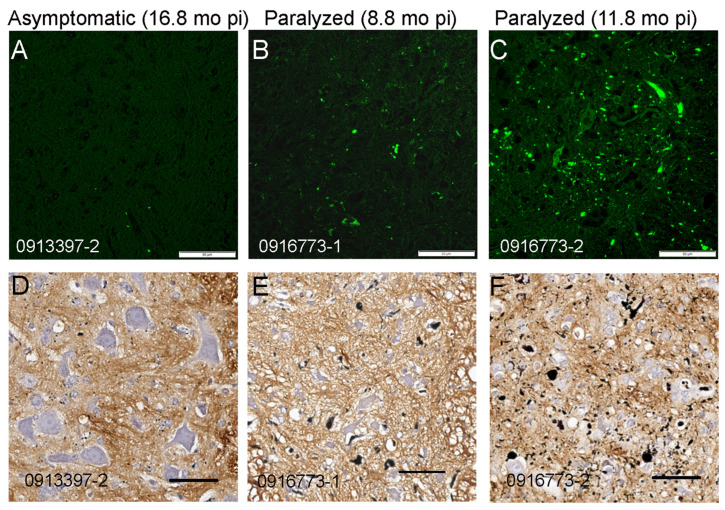
Spinal cords of paralyzed G85R-SOD1:YFP mice exhibit inclusion pathology. (**A**–**C**) Representative images of YFP fluorescence in the lumbar spinal cords of an asymptomatic animal and two paralyzed animals are shown. (**D**–**F**) Representative images of spinal cords from the same mice stained using Campbell–Switzer silver staining with hematoxylin counterstaining. In the sections from paralyzed mice (**E**,**F**), there is disorganized neuropil with shrunken neuronal cell bodies with numerous argentophilic inclusions. Animal IDs are indicated in the bottom left of each panel. Scale bars = 50 µm.

**Figure 3 viruses-15-01819-f003:**
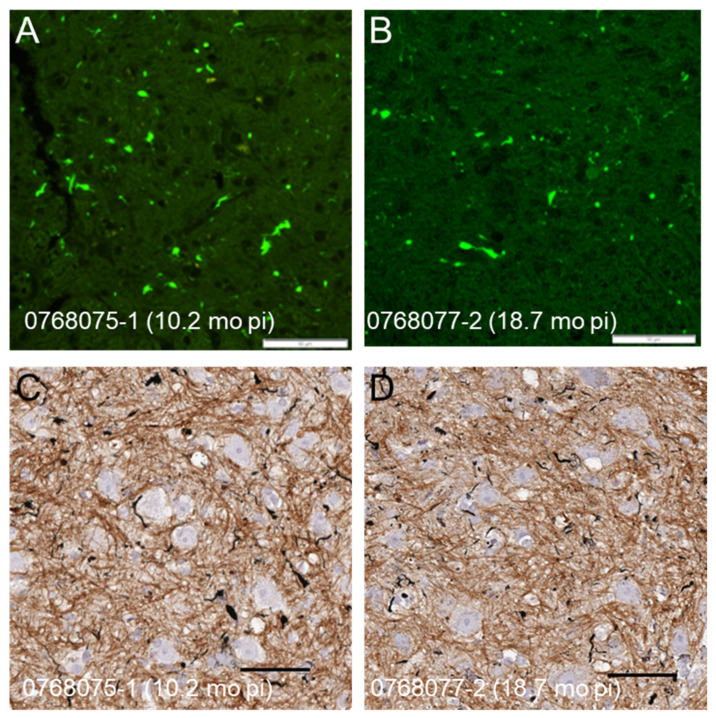
Inclusion pathology in paralyzed G85R-SOD1:YFP mice inoculated with first-passage homogenates of recG93A ALS conformers. (**A**,**B**) Representative images of YFP fluorescence in the lumbar spinal cords of two paralyzed animals are shown. (**C**,**D**) Representative images of spinal cords from the same mice stained via Campbell–Switzer silver staining with hematoxylin counterstaining. Numerous argentophilic inclusions are visible throughout the neuropil. Animal IDs are indicated in the bottom left of each panel. Scale bars = 50 µm.

**Figure 4 viruses-15-01819-f004:**
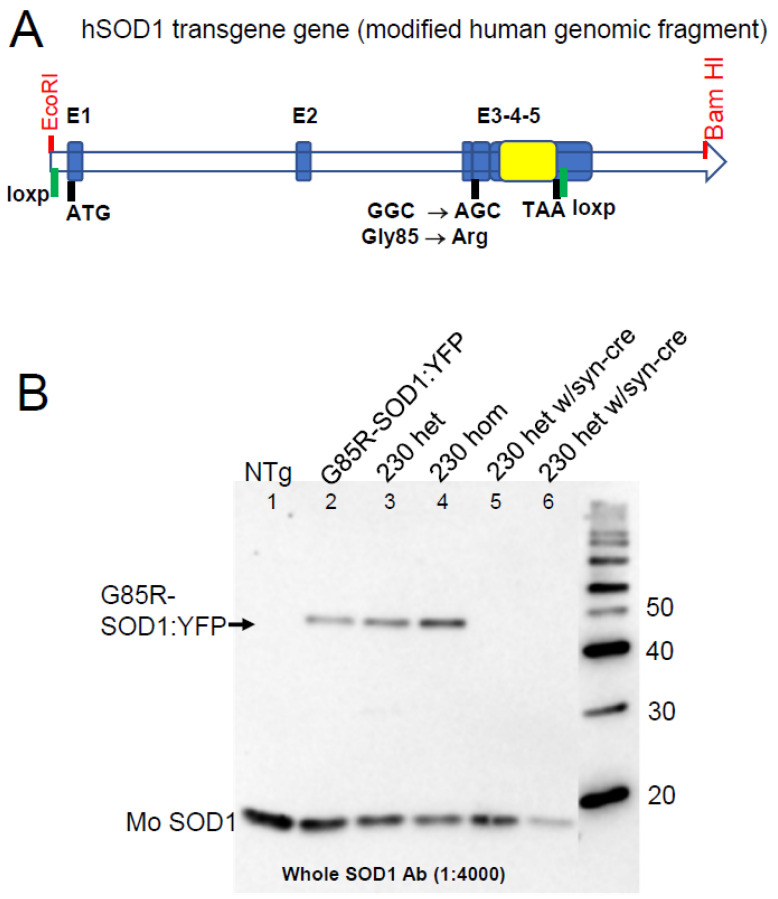
Characterization of loxpG85R-SOD1:YFP mice. (**A**) A schematic diagram of the transgene construct is shown. The starting construct was a version of genomic human SOD1 DNA that had been modified such that exons 3, 4, and 5 were fused, eliminating sequences for introns 3 and 4. The diagram indicates the positions of the YFP cDNA (yellow box), start and stop codons, the G85R mutation, loxp sites, and restriction endonuclease sites to linearize the DNA. (**B**) A representative image of immunoblot analysis of G85R-SOD1:YFP in spinal cord tissues (15 µg total protein per lane) from the original G85R-SOD1:YFP line and mice from Line 230 with the loxpG85R-SOD1:YFP construct. The antibody was a rabbit polyclonal antibody raised against the human SOD1 protein that recognizes both human and mouse SOD1 [29,30]. Mice that are transgenic for Cre recombinase (synapsin promoter) and loxpG85R-SOD1:YFP have undetectable levels of transgene expression. Endogenous mouse SOD1 is visible in all lanes. The image was digitally manipulated to remove lanes after lane 6 and align the lane containing molecular weight standards.

**Figure 5 viruses-15-01819-f005:**
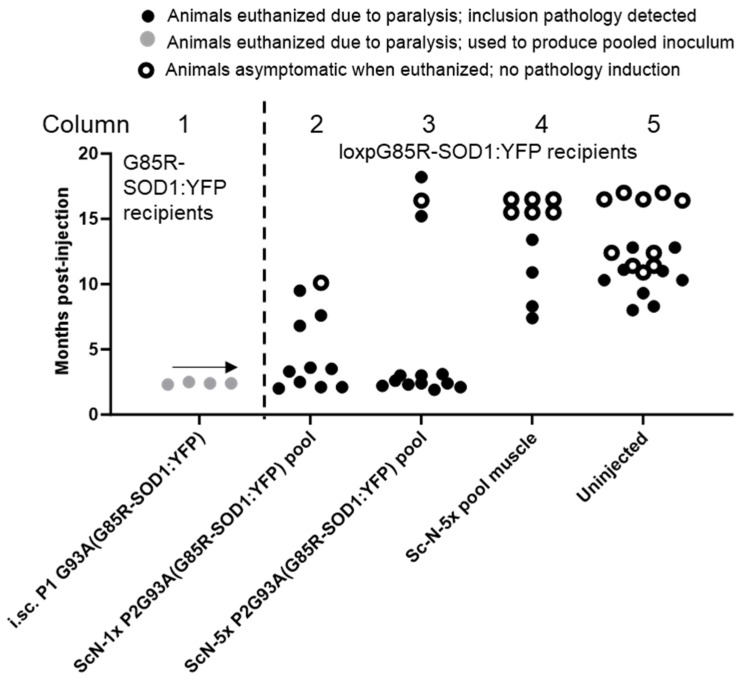
Accelerated paralysis in loxpG85R-SOD1:YFP Line 230 mice injected with tissue homogenates containing G93A-SOD1 ALS conformers. Column 1 shows the age to paralysis for animals that were selected to produce a pooled inoculum from banked tissues. This material was used as inoculum for sciatic nerve injections in the loxpG85R-SOD1:YFP mice (Column 2). A more concentrated preparation of this inoculum was injected into the sciatic nerve of a second cohort (Column 3) and into the muscle (Column 4). The ages to paralysis or harvest of a cohort of uninjected loxpG85R-SOD1:YFP are shown in Column 5.

**Figure 6 viruses-15-01819-f006:**
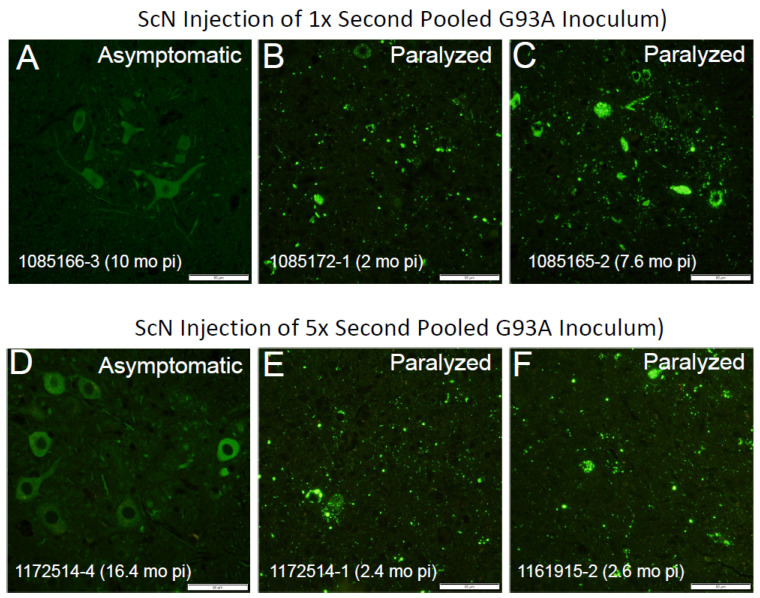
Inclusion pathology in paralyzed loxpG85R-SOD1:YFP mice. (**A**–**C**) Representative images of YFP fluorescence in the lumbar spinal cords of an asymptomatic animal and two paralyzed animals are shown. These examples are from mice injected with the unconcentrated pooled G93A inoculum. (**D**–**F**) Representative images of spinal cords from loxpG85R-SOD1:YFP mice injected with the concentrated inoculum. Animal IDs are indicated in the bottom left of each panel. Scale bars = 50 µm.

**Figure 7 viruses-15-01819-f007:**
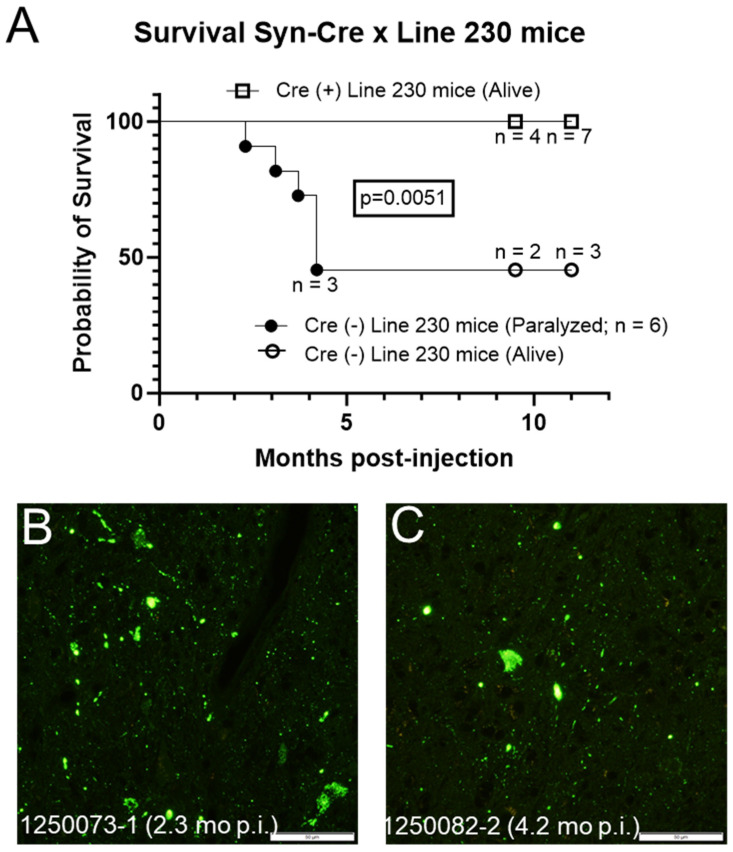
Neuronal deletion of the loxpG85R-SOD1:YFP transgene dramatically reduces susceptibility to inoculum containing G93A-SOD1 ALS conformers. (**A**) Survival plots for loxpG85R-SOD1:YFP (Line 230 mice) that are either positive or negative for the Syn-Cre transgene. At the time of writing, all 11 of the Line 230 mice that were Cre(+), were still alive, whereas 6 of 11 Line 230 mice that were Cre(−) had become paralyzed. (**B**,**C**) Representative images of lumbar spinal cords from paralyzed Line 230-Cre(−) mice. Scale bars = 50 µm.

## Data Availability

All data used in this study are provided in the manuscript in the relevant section. There are no underlying data available.

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
