# Peer review of "Multiple Factors Influence the Incubation Period of ALS Prion-like Transmission in SOD1 Transgenic Mice"

_viruses, 2023, doi:10.3390/v15091819_

Round 1
Reviewer 1 Report
In this manuscript, the authors attempt to identify factors affecting the incubation period of ALS transmission in SOD1 mice. The authors investigated different factors, including the source of the inoculum, the route of infection and the expression of the protein on different cell types. More specifically, they used diverse G93A-SOD1 isolates, originating from one or more terminally ill mice or of recombinant origin, they injected the inocula into the spinal cord, sciatic nerve or hind-limb muscle and used transgenic mice with constitutive expression of the human G85R-SOD1, or expression limited to non-neuronal cells.
The authors report that the source of the inoculum can introduce significant variability to the incubation period of the mice, that can be reduced through serial passage, while the route of administration does not affect the minimum incubation period to paralysis. The authors also indicate that expression of the human mutant SOD protein in the neurons is required for efficient propagation.
The manuscript is well-organised and clearly written. The results are presented in a logical continuum and are easy to follow and the conclusions are supported by the results.
I only have some minor points of criticism, that have also been raised by the authors in the "Discussion" section. The first concerns the use of the recombinant fibrils: I would expect the recombinant fibrils to be much more consistent in structure and seeding properties and maybe differences in the host are responsible for the deviation in incubation period. Maybe the authors could comment more on that.
The second minor point of criticism (also raised by the authors in the "Discussion") is about the technical aspect of the study and how it may affect the outcome. In order to reduce the variability of the incubation period, the authors tried a concentrated inoculum. Maybe in addition to a concentrated inoculum, a diluted one, carrying the same "infectivity" units should also be tried. In this case minute differences in the volumes of inoculum injected would have a smaller effect on the incubation period. Also they could try inocula after more serial passages to stabilise the strain and facilitate identification of the factors affecting propagation.
On a different note, I think that moving Figure 5 before Figure 6 would facilitate reading the manuscript.
Author Response
Comment: I only have some minor points of criticism, that have also been raised by the authors in the "Discussion" section. The first concerns the use of the recombinant fibrils: I would expect the recombinant fibrils to be much more consistent in structure and seeding properties and maybe differences in the host are responsible for the deviation in incubation period. Maybe the authors could comment more on that.
Response: We also thought that recombinant fibrils would produce more consistent results; however our experience has been that they do not. We previously published a study that examined the performance of recombinant fibrils in inducing paralysis in G85R-SOD1:YFP mice (Ayers 2021, PMID: 34016165 ). Several different mutant variants were tested in P0 i.sc. injection and all were found to induce paralysis, but the incubation periods were variable. Here we selected tissue from a G85R-SOD1:YFP animal that developed paralysis after injection of recombinant G93A fibrils at the earliest age for the cohort, which in this case was about 8 months. Although this inoculum induced paralysis at a high frequency (9 of 12 mice injected by ScN route), the incubation period was variable and much longer than the ~2.5 month incubation periods observed in our index study of G93A ALS conformer derived from paralyzed G93A mice. We have made modifications to the text of both Results and Discussion to try and address the reviewer’s comments.
Comment: The second minor point of criticism (also raised by the authors in the "Discussion") is about the technical aspect of the study and how it may affect the outcome. In order to reduce the variability of the incubation period, the authors tried a concentrated inoculum. Maybe in addition to a concentrated inoculum, a diluted one, carrying the same "infectivity" units should also be tried. In this case minute differences in the volumes of inoculum injected would have a smaller effect on the incubation period. Also they could try inocula after more serial passages to stabilise the strain and facilitate identification of the factors affecting propagation.
Response: In the injections in the sciatic nerve, we are somewhat constrained by the small target size of the injection site. Injecting a larger volume is not really feasible. Diluting the inoculum would mean less infectivity would be injected. The purpose of the more concentrated inoculum was to see if we were injecting a saturating level of “infectivity” to achieve the shortest incubation period. We have reworded the description of these studies to be more clear.
Studies to serially passage the SOD1 ALS strains are ongoing, and beyond the scope of this study.
Comment - On a different note, I think that moving Figure 5 before Figure 6 would facilitate reading the manuscript.
Response - We agree. It was not our intent to have Fig 6 before Fig 5.
Reviewer 2 Report
In the current work, authors investigated the propagation of different isolates of G93A-SOD1 ALS conformers, using a paradigm involving transmission to mice expressing human G85R-SOD1 fused to yellow fluorescent protein (G85R-SOD1:YFP). In these studies, they also utilized a newly developed line of mice in which the G85R-SOD1:YFP construct was flanked by loxp sites, allowing its temporal and spatial regulation. This manuscript can be accepted after major revision.
Here are the points:
-The title is not appropriate for this study as it does not barely reflect the incubation period of ALS SOD1 mice.
-The important role of SOD1 in ALS must be explained in detail in Introduction.
-It was not defined well that G93A ALS conformers propagate primarily in neurons (are there any observations in glial cells?).
-It is not clear how many newborn mice were used for seeding experiments.
-“consider” must be “considered” in the last sentence of “Statistical Analysis”.
-The references are not the latest ones. In particular, authors could cite the recent review studies such as “Int. J. Mol. Sci. 2022, 23, 2400. https://doi.org/10.3390/ijms23052400”
Minor editing of English language required.
Author Response
Comment: -The title is not appropriate for this study as it does not barely
reflect the incubation period of ALS SOD1 mice.
Response- We are not exactly sure what the reviewer means. We have modified the title to be more clear that we are examining prion-like behaviors in SOD1 mice.
Comment: -The important role of SOD1 in ALS must be explained in detail in Introduction.
Response – The first 2 paragraphs of the Introduction have been extensively modified to add more details.
Comment: -It was not defined well that G93A ALS conformers propagate primarily in neurons (are there any observations in glial cells?).
Response – We have modified the Discussion to include what is known about non-neuronal cells in SOD1 models. The reviewer is correct in that we don’t really address whether there is propagation in non-neuronal cells. To be more precise, we have altered the language to indicate the need for neuronal propagation in inducing ALS like paralysis.
Comment: -It is not clear how many newborn mice were used for seeding experiments.
Response – We apologize for not being clear about the number of newborn mice that were used in these studies. There were no new P0 injections for these studies. We were trying to show the history of passage used in generating inoculum. Tissues that were used to prepare inoculum were from tissue banks of previously described animals. We have modified the text in Methods, Figure 5, and the legends of Figures 1 and 5 to make it clear that the tissues from mice that received intraspinal injections as newborn were banked tissue from animals described in prior studies.
Comment: -“consider” must be “considered” in the last sentence of “Statistical Analysis”.
Response – thank you for identifying the typo.
Comment: -The references are not the latest ones. In particular, authors could cite the recent review studies such as “Int. J. Mol.Sci. 2022, 23, 2400. https://doi.org/10.3390/ijms23052400”
Response – The review suggested is not really relevant. It seems to be a good review of drug development in ALS, but it does not really review literature related to SOD1 ALS in much detail, other that use of G93A mice in drug discovery.
Reviewer 3 Report
Ayers et al provide analyses of factors influencing the transmission or prion-like behavior of ALS-associated SOD1 aggregates. Having previously reported that some such aggregates, whether from human patients or synthetic SOD1 assemblies can induce a spreading paralysis in Tg mouse models, they now extend their analyses to other aberrant SOD1 conformers when injected into Tg mice, including a new Tg line that allowed the preferential reduction of expression of SOD1 in neurons. Understanding factors that control the efficiencies of induction and rates of spreading of pathological SOD1 aggregation is highly relevant to the understanding of ALS pathogenesis. Although a bit complicated to follow, the experiments are well-conceived and explained, for the most part. The results of the various transmission experiments are a bit variable in terms of incubation periods and attack rate (this is new-fangled biology, after all), but they reveal significant observations with respect to differences between ex vivo and synthetic recombinant aggregates, as well as comparisons of inoculations into the spinal cord, sciatic nerve, and mice with little neuronal SOD1 expression. I have only a few minor suggestions for improvement.
1) Define MND on first use, or spell it out as done in most of the manuscript.
2) The manuscript should be checked for redundancies, e.g. “developed paralysis at 5.5 months of age” in the first paragraph of Results.
3) Does coexpression of mouse SOD1 affect human SOD1 aggregation? It would be helpful if this possibility were openly considered.
Author Response
Comment - Define MND on first use, or spell it out as done in most of the manuscript.
Reply - corrected
Comment - The manuscript should be checked for redundancies, e.g. “developed paralysis at 5.5 months of age” in the first paragraph of Results.
Reply – Thanks for catching this. Corrected
Comment - Does coexpression of mouse SOD1 affect human SOD1 aggregation? It would be helpful if this possibility were openly considered.
Reply – The potential for mouse SOD1 to interact with or affect SOD1 aggregation has been addressed in several studies. The most definitive study was that of Bruijn et al (Science 1998, 281:1851–1854) in which mice expressing human G85R SOD1 were crossed to SOD1 knockout mice. Deleting mouse SOD1 had no impact on disease onset or the level of mutant human SOD1 aggregation. Additionally, mouse SOD1 is not present in detergent-insoluble fractions containing the misfolded human SOD1 aggregates Jonsson et al (Brain. 2004;127:73–88).
Round 2
Reviewer 2 Report
The authors have satisfactorily addressed most of my concerns and thus, the paper is acceptable for publication.
Minor editing of English language required.